# Ergosta-7, 9 (11), 22-trien-3β-ol Interferes with LPS Docking to LBP, CD14, and TLR4/MD-2 Co-Receptors to Attenuate the NF-κB Inflammatory Pathway *In Vitro* and *Drosophila*

**DOI:** 10.3390/ijms22126511

**Published:** 2021-06-17

**Authors:** Wen-Tsong Hsieh, Min-Hsien Hsu, Wen-Jen Lin, Yi-Cheng Xiao, Ping-Chiang Lyu, Yi-Chung Liu, Wei-Yong Lin, Yueh-Hsiung Kuo, Jing-Gung Chung

**Affiliations:** 1Department of Pharmacology, China Medical University, Taichung 40402, Taiwan; 2Chinese Medicine Research Center, China Medical University, Taichung 40402, Taiwan; kuoyh@mail.cmu.edu.tw; 3Department of Neurology, Chang Bing Show-Chwan Memorial Hospital, Changhua 505, Taiwan; roderick009@gmail.com; 4Graduate Institute of Biomedicine Science, China Medical University, Taichung 40402, Taiwan; u104001424@cmu.edu.tw; 5School of Medicine, College of Medicine, National Cheng Kung University, Tainan 701, Taiwan; petermq3386@gmail.com; 6Institute of Bioinformatics and Structural Biology, National Tsing-Hua University, Hsinchu 300044, Taiwan; pclyu@mx.nthu.edu.tw; 7Institute of Population Health Sciences, National Health Research Institutes, Miaoli 350, Taiwan; jong212@gmail.com; 8Graduate Institute of Integrated Medicine, College of Chinese Medicine, China Medical University, Taichung 40402, Taiwan; linwy@mail.cmu.edu.tw; 9Department of Chinese Pharmaceutical Sciences and Chinese Medicine Resources, China Medical University, Taichung 40402, Taiwan; 10Department of Biological Science and Technology, China Medical University, Taichung 40402, Taiwan; jgchung@mail.cmu.edu.tw

**Keywords:** Ergosta-7, 9 (11), 22-trien-3β-ol (EK100), LPS, LBP, CD14, TLR4/MD-2, NF-κB, *Drosophila*

## Abstract

Ergosta-7, 9 (11), 22-trien-3β-ol (EK100) was isolated from *Cordyceps militaris*, which has been used as a traditional anti-inflammatory medicine. EK100 has been reported to attenuate inflammatory diseases, but its anti-inflammatory mechanism is still unclear. We were the first to investigate the effect of EK100 on the Toll-like receptor 4 (TLR4)/nuclear factor of the κ light chain enhancer of B cells (NF-κB) signaling in the lipopolysaccharide (LPS)-stimulated RAW264.7 cells and the green fluorescent protein (GFP)-labeled NF-κB reporter gene of *Drosophila*. EK100 suppressed the release of the cytokine and attenuated the mRNA and protein expression of pro-inflammatory mediators. EK100 inhibited the inhibitor kappa B (IκB)/NF-κB signaling pathway. EK100 also inhibited phosphatidylinositol-3-kinase (PI3K)/Protein kinase B (Akt) signal transduction. Moreover, EK100 interfered with LPS docking to the LPS-binding protein (LBP), transferred to the cluster of differentiation 14 (CD14), and bonded to TLR4/myeloid differentiation-2 (MD-2) co-receptors. Compared with the TLR4 antagonist, resatorvid (CLI-095), and dexamethasone (Dexa), EK100 suppressed the TLR4/AKT signaling pathway. In addition, we also confirmed that EK100 attenuated the GFP-labeled NF-κB reporter gene expression in *Drosophila*. In summary, EK100 might alter LPS docking to LBP, CD14, and TLR4/MD-2 co-receptors, and then it suppresses the TLR4/NF-κB inflammatory pathway in LPS-stimulated RAW264.7 cells and *Drosophila*.

## 1. Introduction

Chronic inflammatory diseases are the most significant cause of death in the world. Nearly 60% of people die due to chronic inflammatory diseases like stroke, chronic respiratory diseases, heart disorders, cancer, obesity, and diabetes [1]. In most inflammatory disorders, the lipopolysaccharide (LPS) innate binds to Toll-like receptor 4 (TLR4) co-receptors to activate inflammatory and immune responses [2]. TLR4/Myeloid differentiation factor 88 (MyD88) signaling could activate phosphatidylinositol-3-kinase (PI3K) and Protein kinase B (Akt), then it could stimulate the phosphorylates IκB kinase (IKK) and inhibitor of kappa B (IκB) pathway to promote the nuclear factor of the κ light chain enhancer of B cells (NF-κB) inflammatory signaling [3]. The NF-κB family is a multifunctional transcription factor that plays a vital role in initiating and developing inflammation [4]. In general, it always stabilizes NF-κB in the cytoplasm by binding to the IκB junction [5]. Moreover, IκB phosphorylation is a critical step in activating NF-κB [6]. However, this step can promote NF-κB p65 (p65) and NF-κB p50 (p50) translocation from the cytoplasm to the cellular nucleus [7]. The cellular NF-κB regulates gene expression by binding to regions of DNA, then it promotes the creation of inflammatory cytokines such as interleukin (IL)-1, IL-6, IL-8, and tumor necrosis factor (TNF)-α [8]. The suppression of the TLR4/MyD88/NF-κB signaling pathway may be an effective strategy to prevent or reduce chronic inflammatory diseases [9].

Ergosta-7, 9 (11), 22-trien-3β-ol (EK100) (Figure 1A) was isolated from the *Cordyceps militaris,* which has been extensively used as traditional anti-inflammatory medicine [10]. EK100 could attenuate hypertriglyceridemia, hypercholesterolemia, and hepatic fat accumulation in high-fat diet-fed mice [11]. EK100 has attenuated the translation of various pro-inflammatory genes, which interfered with the inflammatory and immune responses [12]. EK100 reduced p65 expression via inhibited PI3K/Akt-associated glycogen synthase kinase 3 (GSK3) and β-catenin activation in ischemic brain injury mice [13]. EK100 reduces the expression of inducible nitric oxide synthase (iNOS) and cyclooxygenase-2 (COX-2) [14]. EK100 also decreases inflammatory factor expression NF-κB and inflammatory cytokines IL-6 and TNF-α in mice [15,16]. All reports have indicated that more upstream anti-inflammatory mechanisms of EK100 could be possible. No reports in the available literature have revealed EK100 effects on TLR4 relative signaling. In the present study, we first investigated the anti-inflammatory effects of EK100 on influencing TLR4/NF-κB relative inflammatory signaling in LPS-stimulated RAW264.7 cells.

## 2. Results

### 2.1. Effects of EK100 on the Cell Viability and Cytokines Release in LPS-Stimulated RAW264.7 Murine Macrophage Cells

A 3-(4,5-Dimethylthiazol-2-yl)-2,5-diphenyltetrazolium bromide (MTT) reagent was used to evaluate the cell viability. EK100 under 80 μM showed no significant effects on cell viability (Figure 1B). EK100 inhibited the nitrite production of LPS stimulated in a dose-dependent manner (Figure 1C). Moreover, EK100 decreased PGE_2_ and IL-1β release in LPS-induced RAW264.7 cells (Figure 1C–D). Our results demonstrated that 80 μM of EK100 inhibited the formation of the inflammatory cytokines of NO to 2.15 ± 0.20 μM compared to the LPS alone group (19.8 ± 0.8 μM) (Figure 1C), of PGE_2_ to 110.0 ± 10.2 pg/mL as compared to the LPS alone group (161.8 ± 9.1 pg/mL) (Figure 1D), and of IL-1β to 1.1 ± 0.1 pg/mL as compared to the LPS alone group (9.1 ± 0.1 pg/mL) (Figure 1e), respectively. In summary, EK100 could suppress the production of serval inflammatory cytokines in the LPS-induced RAW264.7 cells.

### 2.2. EK100 Suppressed LPS-Induced mRNA and Protein Expression of Pro-Inflammatory Mediators of iNOS and COX-2 in RAW264.7 Cells

LPS could increase the protein expression of pro-inflammatory mediators, iNOS and COX-2. The maximum period of protein expression presented in 12–24 h (Figure 2A). In Figure 2B, 80 μM of EK100 suppressed the protein expression of iNOS and COX-2 in LPS-stimulated RAW264.7 cells for 24 h, respectively. As shown in Figure 2B, quantitative analysis revealed that EK100 (80 μM) decreased the protein expression of iNOS to 0.43 ± 0.03 fold and of COX-2 to 0.07 ± 0.02 fold, as compared to the LPS alone group. Moreover, EK100 also decreased the mRNA levels of iNOS and COX-2 in LPS-stimulated RAW264.7 cells for 6 h, respectively. As shown in Figure 2C, 80 μM of EK100 inhibited the mRNA expression of iNOS to 0.43 ± 0.10 fold and COX-2 to 0.17 ± 0.03 fold, respectively. The results showed that EK100 suppressed LPS-induced mRNA and protein expression of pro-inflammatory mediators of iNOS and COX-2 in RAW264.7 cells.

### 2.3. Effects of EK100 on the Phosphorylation of IKK/IκB and the Translocation of NF-κB in LPS-Induced RAW264.7 Cells

As shown in Figure 3A, we found that LPS increased the phosphorylation of IKK and IκB, with the maximum peak being reached within 15–30 min. EK100 inhibited LPS-stimulated phosphorylation of IKK and IκB in a dose-dependent manner. EK100 at 80 μM inhibited the p-IKK to 0.09 ± 0.01 fold and the p-IκB to 0.48 ± 0.01 fold, compared to the LPS alone group, respectively (Figure 3B). Moreover, EK100 inhibited the LPS-stimulated nuclear translocation factor of p65 and p50 in RAW264.7 cells. EK100 at 80 μM inhibited the nuclear protein expression of the translocation factor of p65 to 0.27 ± 0.03 fold and of p50 to 0.44 ± 0.04 fold, respectively (Figure 3C). Furthermore, EK100 inhibited the nuclear factor, NF-κB, bonded to DNA to develop the DNA–NF-κB complex dose-dependently with the electrophoretic mobility shift assay (EMSA) (Figure 3D). Additionally, EK100 suppressed the translocation of p65 to the nucleus in LPS-stimulated RAW264.7 murine macrophages in relation to the immunofluorescence analysis (Figure 3E). In summary, we suggest that EK100 suppressed LPS-induced phosphorylation of IKK/IκB and the translocation of NF-κB in RAW264.7 cells.

### 2.4. Effects of EK100 on the PI3K/Akt Signaling Pathway in LPS-Induced RAW264.7 Cells

As shown in Figure 4A, we found that EK100, induced dose-dependently, inhibited LPS-stimulated phosphorylation of PI3K and Akt, respectively. Compared with dexamethasone (Dexa) (2 μM), EK100 (80 μM) inhibited the phosphorylation of PI3K and AKT, with the expression of p-PI3K decreased to 0.13 ± 0.01 fold and p-AKT decreased to 0.06 ± 0.01 fold compared to the LPS alone group, respectively (Figure 4B). As shown in Figure 4B, compared with TLR4 inhibitor, resatorvid (CLI-095) (15 μM), EK100 inhibited the phosphorylation of IKK and IκB, with the expression of p-IKK decreased to 0.25 ± 0.01 fold and p-IκB decreased to 0.64 ± 0.03 fold, compared to the LPS alone group, respectively. As shown in Figure 4C, compared with CLI-095 (15 μM), EK100 inhibited the phosphorylation of AKT, with the expression of p-AKT decreased to 0.57 ± 0.04 fold compared to the LPS alone group, respectively. As shown in Figure 4D, compared with CLI-095 (15 μM), EK100 inhibited the expression of iNOS, and it decreased to 0.04 ± 0.01 fold, and COX-2 decreased to 0.10 ± 0.01 fold, compared to the LPS alone group, respectively. In summary, EK100 attenuated LPS-induced phosphorylation of PI3K/Akt and activated the TLR4/MyD88 signal pathway in RAW264.7 cells.

### 2.5. Effects of EK100 on the Protein Expression of Lipopolysaccharide-Binding Protein (LBP), the Cluster of Differentiation 14 (CD14), and TLR4/Myeloid Differentiation-2 (MD-2) Co-Receptors in LPS-Stimulated RAW264.7 Cells

This study also aimed to identify the protein on the LPS/TLR4 signaling pathway. In comparison with the LPS alone group (100%), EK100 (80 μM) inhibited the LPS-stimulated protein expressions of LBP to 21.35%, CD14 to 21.31%, and TLR4 to 38.78%, respectively (Figure 5A). Dexa (2 μM) also inhibited the protein expression of LBP to 14.91%, CD14 to 8.54%, and TLR4 to 39.21%, respectively (Figure 5A). Moreover, EK100 and Dexa suppressed the expressions of LBP, CD14, and TLR4 in a dose-dependent manner. However, the images of immunofluorescence (IF) staining assays showed that EK100 and Dexa decreased the LPS-binding of relative proteins of TLR4 co-receptor expression of LBP, CD14, MD-2, and TLR4 (Figure 5B–E). The results suggested that EK100 and Dexa interfered with the LPS binding to LBP, CD14, and MD-2 TLR4 co-receptors.

### 2.6. Effects of EK100 on LPS Binding to LBP, CD14, and TLR4/MD-2 with Protein–Ligand Docking

We simulated EK100, and Dexa interfered with LPS binding to LBP, CD14, the TLR4/MD-2 complex co-receptor within the LigPlot+ 2D diagrams protein–ligand docking model. EK100 had hydrophobic interactions with LBP, including residues His79, Lys115, Trp116, Leu125, His126, and Gly127 tight binding with LBP. Simulation results indicated that the best binding affinity of EK100 was −6.8 kcal/mol, and of Dexa, it was −6.2 kcal/mol, respectively (Figure 6A). EK100 had hydrophobic interactions in the residues of Val57, Val73, Leu89, Val91, Val101, Leu105, Leu106, and Ala10,9 and it has a hydrogen bond in Ser53 to bind to CD14. Theoretical analysis and computer simulation results showed that the best binding affinities of EK100 were −9.3 kcal/mol, and of Dexa, it was −7.6 kcal/mol, respectively (Figure 6B). EK100 also had a hydrophobic interaction with residues Ile44, Tyr65, Leu71, Leu74, Phe76, and Phe147. The results demonstrated that EK100 and Dexa could significantly improve the binding affinities in the hydrophobic pocket of MD-2, regarding PyRx docking, which were found to be −11.3 and −8.5 kcal/mol, respectively (Figure 6C). In conclusion, the simulation results indicated that EK100 and Dexa could interfere with the receptor binding of LBP, CD14, and MD-2 on the TLR4 co-receptors in LPS-stimulated RAW264.7 cells.

### 2.7. Effects of EK100 on the Green Fluorescent Protein (GFP)-Labeled Rel/NF-κB Fluorescence Expression in Drosophila

To investigate the effect of EK100 on the TLR4/NF-κ inflammatory pathway, the expression level of NF-κB, a specific marker gene of the TLR4/NF-κ inflammatory pathway, was quantified. Results showed that NF-κB protein expression level was quantified by measuring the fluorescence of GFP-labeled Rel/NF-κB in *Drosophila* (Figure 7A). Additionally, EK100 could suppress LPS-stimulated NF-κB protein expression level, compared with indomethacin (Figure 7B). Overall, EK100 could inhibit the GFP-labeled Rel/NF-κB fluorescence expression in *Drosophila*. 

## 3. Discussion

TLR4 activates the inflammatory and immune responses and defends exogenous or endogenous exposures [17]. Moreover, LPS/TLR4 activates MyD88-dependent signaling, stimulates the NF-κB pathway, and releases cytokines [18]. TLR4 co-receptors activates the MyD88 and the NF-κB signaling pathways and enhance COX-2 expression [19]. However, NF-κB translocated from the cytoplasm to the nucleus produces a diversity of inflammatory cytokines, including NO, PGE_2_, and IL-1β [20]. However, LPS/TLR4-mediated stimulation of PGE_2_ production was blocked by a selective COX-2 inhibitor [21]. NF-κB signal pathways affect pro-inflammatory mediators, such as iNOS and COX-2, and inflammatory cytokines, including NO, PGE_2_, and interleukin-1β (IL-1β) [22]. Herein, EK100 under 80 μM showed no significant effects on cell viability (Figure 1B). EK100 suppressed the release of cytokines NO, PGE_2_, and IL-1β in LPS-stimulated RAW264.7 cells (Figure 1B–D). Moreover, EK100 inhibited the mRNA and protein expression of LPS-stimulated pro-inflammatory mediators iNOS and COX-2 in LPS-stimulated RAW264.7 cells (Figure 2b,c). In the present study, EK100 decreased the release of cytokines such as NO, PGE2, and IL-1β (Figure 1B–D) and the levels of pro-inflammatory mediator proteins iNOS and COX-2 (Figure 2B–C). EK100 might inhibit the inflammatory effect in LPS-stimulated RAW264.7 cells.

LPS/TLR4 activates the MyD88-dependent signaling pathway and stimulates the phosphorylation of IKK [23]. An association with inhibitor IκB proteins regulates Rel/NF-κB transcription factors. The IKK complex could facilitate phosphorylation-induced and proteasome-mediated degradation of IκB to activate and release Rel/NF-κB transcription factors [24]. LPS-induced degradation of IκBα is the leading nuclear factor of p65/p50 translocation and DNA binding [25]. Regarding the results, LPS increased the phosphorylation of IKK and IκB within 15–30 min (Figure 3A). EK100 inhibited LPS-stimulated phosphorylation of IKK and IκB dose-dependently (Figure 3B) and suppressed the protein expression and translocation of p65 and p50 in the LPS-stimulated RAW264.7 cells (Figure 3C). According to these findings above, EK100 might reduce IKK/IκB inflammatory signaling and suppress the p65 and p50 translocation to the nucleus in LPS-stimulated RAW264.7 cells.

The approach of a TLR4 signaling inhibitor is a diverse strategy for the handling of inflammatory diseases [26]. However, CLI-095 and Dexa suppressed LPS-induced transcription of the TLR4/MD-2 co-receptor signaling [27,28]. The following results show that EK100 has anti-inflammatory activity by modulating the TLR4/MyD88 signaling pathways. PI3K/Akt mediated regulation of IKK and NF-κB activation. IKK phosphorylates both the IκB protein and the Rel/NF-κB and enhances the NF-κB transcription factor [29]. TLR4 induced a dose-dependent stimulation of NF-κB activity, which was abrogated in the presence of the TLR4 inhibitor, CLI-095 [30]. Compared with Dexa, EK100 induced dose-dependently inhibited LPS-stimulated phosphorylation of PI3K and Akt (Figure 4A). Compared with the TLR4 inhibitor, CLI-095 (15 μM), EK100 inhibited the phosphorylations of IKK and IκB (Figure 4B), and EK100 inhibited the expression of MyD88, p-AKT, iNOS, and COX-2 expression (Figure 4C–D). This study suggests that EK100 attenuates LPS-induced phosphorylation of PI3K/Akt and EK100 modulation of the TLR4/MyD88 inflammatory signaling pathways in RAW264.7 cells.

In the inflammatory process, LBP, a protein that binds to LPS and transfers LPS monomers to CD14, is driven by circulating concentrations of LPS [31]. However, LPS binding to LBP and the LBP transfer of LPS to mCD14 are necessary for the TLR4/MD-2 internalization and for prompting the inflammatory signaling pathway [32]. The results prove that EK100 suppressed the LPS-relative binding protein expressions of LBP, CD14, MD-2, TLR4, and MyD88 in LPS-stimulated RAW264.7 cells (Figure 5A–E). The results indicated that EK100 and Dexa interfered with the LPS binding to LBP, CD14, and MD-2 TLR4 co-receptors.

This study aimed to unravel the atomic details regarding the molecular mechanism of receptors recognition and the ligand–receptor interactions by applying molecular modeling. The computational protein–ligand docking methods can help develop novel drugs for TLR4-related disorders [26]. The N-terminal domain of LBP contains conserved residues with either positively charged or moderately hydrophobic side chains, and mutations of residues Arg119, Lys120, and Lys124 in human LBP abolish LPS binding [33]. The EK100–LBP docking result showed that EK100 interacts with Arg119 and Lys124 of LBP through hydrophobic interactions. The above experimental results also confirm that EK100 competes with LPS for binding to LBP (Figure 6A). The binding affinity of EK100 in CD14 includes hydrophobic interactions (including the identical residues mentioned above, such as Val52, Ser53, and Leu89) and the Ser53 hydrogen bond interaction. Theoretical analysis and computer simulation results can explain the impact of EK100, which may have other hydrophobic interactions with CD14 (Figure 6B). LPS, translocated to the complexing myeloid differentiation-2 (MD-2), activates the TLR4 receptor co-receptor signaling pathway [34]. Moreover, MD-2 is physically associated with TLR4 on the cell surface and recognizes LPS to elicit an innate immune response [35]. The comparison showed that EK100 also has a hydrophobic interaction with TLR4/MD-2 co-receptor, and it was found to be in exceptional agreement with the molecular aspects of TLR4 activation and signaling by computational approaches (Figure 6C). The results indicate that EK100 could interfere with LPS binding to LBP, CD14, and the MD-2/TLR4 co-receptor on the cellular membrane surface and modulate the TLR4/MyD88 inflammatory signaling pathways in RAW264.7 cells.

Recently, TLR4 therapy has been proposed to be another aspect of anti-inflammation [36]. Rel/NF-κB transcription factors form an integral part of the inflammatory and innate immune defense system [37]. Moreover, Rel/NF-κB pathways play a significant role in *Drosophila* host defense [38]. The results showed that EK100 attenuated the NF-κB protein expression level, and this was quantified by measuring the fluorescence of GFP-labeled Rel/NF-κB in *Drosophila* (Figure 7A). Additionally, EK100 could suppress the LPS-stimulated NF-κB protein expression level compared with indomethacin (Figure 7B). Overall, EK100 could inhibit the GFP-labeled Rel/NF-κB fluorescence expression in *Drosophila*.

## 4. Materials and Methods

### 4.1. Materials

EK100 was isolated from *Cordyceps militaris*. Dimethyl sulfoxide (DMSO), LPS (Lipopolysaccharides), MTT reagent, and Griess reagent were purchased from Sigma-Aldrich (Louis, MO, USA). Dulbecco’s Modified Eagle Medium (DMEM), fetal bovine serum (FBS), 100 U/mL penicillin, 100 μg/mL streptomycin, Trypsin-EDTA, TRIzol™ Reagent, SuperScript™ II Reverse Transcriptase, RNaseOUT™ Recombinant RNase Inhibitor, Hoechst 33258, Alexa Fluor 488, and Alexa Fluor 594 were purchased from Invitrogen (Carlsbad, CA, USA). The primary antibodies, anti-iNOS, COX-2, p65, and p50, were obtained from Santa Cruz (Menlo Park, CA, USA). The primary antibodies, anti-Akt, p-Akt, p-IKK, IKK-α, IKK-β, p-IκB, and IκB, were obtained from Cell Signaling (Beverly, MA, USA). A prostaglandin E_2_ ELISA Kit was purchased from Cayman (Ann Arbor, MI, USA).

### 4.2. Cell Line and Culture

The RAW264.7 murine macrophage cell line was obtained from the Food Industry Research and Development Institute (Hsinchu, Taiwan). Cells were cultured in DMEM medium supplemented with 10% FBS, 100 units/mL penicillin, and 100 μg/mL streptomycin in an incubator at 37 °C with 5% CO_2_ [39].

### 4.3. Cell Viability Assay

As previously described, the RAW264.7 murine macrophage cell viability was determined using the MTT assay [40]. RAW264.7 cells (3 × 10^4^/well) were seeded in 96-well cell culture plates overnight. Cells were treated with EK100 (0, 10, 20, 40, and 80 μM) (dissolve in under 1% DMSO) for 1 h and then incubated with LPS (100 ng/mL) for 24 h. Then, the culture medium was replaced with 100 μL of 0.5 μM MTT reagent and incubated at 37 °C for 2 h. The culture medium was removed, and 100 µL of 0.04 N HCl/isopropanol was added. The absorbance at 570 nm was measured with an EPOCH2 plate reader (BioTek, Winooski, VT, USA). Optical density (OD) values were used to indicate the levels of cell viability.

### 4.4. Nitrite Assay

In the inflammatory stage, nitrite can cause injury and contribute to oxidative tissue damage [41]. Nitrite production was determined using the Griess reagent as previously described [42]. RAW264.7 cells (3 × 10^4^/well) were seeded overnight in 96-well cell culture plates. Cells were treated with EK100 (0, 10, 20, 40, and 80 μM) for 1 h before being incubated with LPS (100 ng/mL) for 24 h. Cell culture supernatants were collected using the Griess reagent to detect nitrite production and measure the absorbance at 540 nm with the EPOCH2 plate reader.

### 4.5. Enzyme-Linked Immunosorbent Assay (ELISA)

The production of cytokines was detected with an ELISA reader, as previously described [43]. RAW264.7 cells (3 × 10^4^/well) were seeded overnight in 96-well cell culture plates. Cells were treated with EK100 (0, 10, 20, 40, and 80 μM) for 1 h before being incubated with LPS (100 ng/mL) for various periods. The cell culture supernatant was collected and tested for cytokine levels using an ELISA kit (PGE2, IL-6, and TNF-α). It measured the absorbance with an EPOCH2 plate reader.

### 4.6. Quantitative Real-Time Polymerase Chain Reaction (Q-PCR)

The total RNA was extracted using a Trizol reagent kit, and PCR reactions were performed as previously described [44]. RAW264.7 cells (1.5 × 10^5^ cells/well) in 6-well plates were treated with EK100 (0, 10, 20, 40, and 80 μM) for 1 h and with LPS (100 ng/mL) for various periods. Total RNA extracted by the TRIzol reagent was reverse-transcribed to DNA by using the cDNA Reverse Transcription Kit. SYBR Green Master Mix was used to perform PCR reactions by StepOne Plus Real-Time PCR Systems under the following conditions: 95 °C for 10 min, 42 cycles at 95 °C for 10 s, and then 60 °C for 60 s. The following primers were used for iNOS, sense primer, 5′-AGCAACTACTGCTGGTGGT-3′, and antisense primer, 5′-AATGGGCAGACTCTGAAGA-3′; the primers used for COX-2 were sense primer, 5′-CTGGAACATGGACTCACTCAGTTTGT-3′, and antisense primer, 5′-ACAAGCAGTGGCAAAG- GCCT-3′; the primers used for GAPDH were sense primer, 5′-GGCCTTCCGTGTTCCTACC-3′, and antisense primer, 5′-GAAGGTGGTGAAGCAGGCA-3′. The results were expressed as the ratio of the optical density to GAPDH.

### 4.7. Western Blot Assay

Western blotting was used for immunodetection, modification, and quantification of proteins, as described previously [45]. RAW264.7 cells were treated with EK100 and incubated with LPS. The PRO-PREP™ Protein Extraction Solution (iNtRON, Seoul, Korea) was added to extract proteins from cells. The total proteins were separated by SDS-PAGE on 8–12% sodium dodecyl sulfate-polyacrylamide gel (SDS-PAGE) and transferred to Polyvinylidene fluoride (PVDF) membranes. Subsequently, the membranes were blocked with 5% Albumin for 1 h and probed with primary antibodies overnight at 4 °C. The corresponding secondary antibodies were added and conjugated with horseradish peroxidase on the PVDF membranes for 1 h at room temperature. The immunoreactive proteins were detected by enhanced chemiluminescence reagents (Perking Elmer, Waltham, MA, USA). Western blot analysis was quantified using the GE Las4000 Mimi Molecular Imaging System (GE Healthcare Co., Piscataway, NJ, USA). The immunofluorescence data were analyzed with TotalLab gel analysis software (BioSystematica, Devon, UK).

### 4.8. Electrophoretic Mobility Shift Assay (EMSA)

An EMSA is a quick and delicate method to detect protein-nucleic acid interactions, as described previously [46]. RAW264.7 cells were treated with EK100 and incubated with LPS. The EMSA was achieved with Odyssey^®^ IRDye^®^ 700 infrared dye-labeled double-stranded oligonucleotides, coupled with the EMSA buffer kit (Li-COR Bioscience, Lincoln, NE, USA). The NF-κB IRDye^®^ 700 infrared dye-labeled oligonucleotide sequences of the double-stranded DNA probes were used as follows: (sense: 5′- GAG CGT GGG GAT CCC GGG AGT C-3′ and anti-sense: 5′-GAC TCC CGG GAT CCC CAC GCT C-3′), and they were used in the GRK5 gene. The oligos were end-labeled using IRdye 700 and used as probes (Integrated DNA Technologies, Coralville, IA, USA). Nuclear proteins (7 µg) were incubated with IRdye-labeled NF-κB oligonucleotides in the dark for 30 min at room temperature in reaction buffer, and 1 µg of poly(dI-dC)-poly(dIdC) (Li-Cor) was used as a nonspecific competitor. The image was visualized and scanned by the odyssey infrared imaging system (Li-COR Bioscience, Lincoln, NE, USA).

### 4.9. Immunofluorescence Assay

The immunofluorescence assay was applied to detect the protein distribution with confocal spectral microscopy, as described previously [47]. RAW264.7 cells were seeded on glass coverslips overnight and then treated with drugs for 1 h, before being incubated with LPS (100 ng/mL) for various periods. Then, they were treated with cold 4% paraformaldehyde for 20 min and then infiltrated with 0.5% Triton-X 100 for 30 min. Cells were blocked with 5% Bovine serum albumin (BSA) for 1 h and incubated with a primary antibody of p65, p50, c-Jun, c-Fos, and p-STAT3 overnight at 4 °C. The cells were then washed three times and incubated with a secondary antibody labeled with Alexa Fluor-594 for 1 h. A solution of 4′,6-diamidino-2-phenylindole (DAPI) (50 μg/mL) was added for 20 min at 37 °C in darkness. The immunofluorescence images were obtained with an SP2/SP8X Confocal Spectral Microscope (Leica, Wetzlar, Germany).

### 4.10. Fluorescence Microscopy and NF-κB Activity Analysis in Drosophila

The sequencing of the genomic insertion sites determines the splicing patterns downstream of the enhanced green fluorescent protein (EGFP) exon, and it analyzes expression patterns. The Bloomington *Drosophila* Stock Center supplied the *Drosophila* line, BDSC #50836, with GFP-labeled NF-κB [48]. *Drosophilas* were kept in a culture tube containing control or di(2-Ethylhexyl) phthalate (DEHP) culture medium in 25 °C, humidity 20%, 12 h day–night-shift. The whole brains of 5-day-old *Drosophila* and third instar *Drosophila* larvae were removed and placed in a plastic petri dish containing the PBS buffer. It was pretreated with LPS (10, 30, 100, and 300 ng/mL) for 98 h. With a protective ring and cover glass to restrict its activities, the whole body was placed under a fluorescent microscope to observe with the naked eye and capture photos for recording. The brains of *Drosophila* larvae were dissected and immersed in 4% paraformaldehyde for 20 min. Then, the coverslips were mounted the slide with nail polish; they were placed under a fluorescent microscope to observe with the naked eye and take a photo for recording. The GFP-labeled Rel/NF-κB fluorescence expression was detected using a microscope with a GFP fluorescence channel (Nikon BX51 (Nikon, Tokyo, Japan) with a CCD camera (SPOT; Diagnostic Instruments; Sterling Heights, MI, USA).

### 4.11. Protein-Ligand Docking Assay

The PyRx/AutoDock Vina is based on the Lamarckian genetic algorithm and empirical free energy score function [49]. The docking software PyRx version 0.98, together with AutoDock Vina, was used for all docking calculations [50]. Docked complexes were visualized and analyzed using the PyMOL Molecular Graphics System (Ver. 2.3 Schrödinger, Portland, OR, USA), and the interactions between the protein and the ligand were analyzed using the “LIGPLOT” module within the LigPlot+ 2D program (v2.2) [51]. The N-terminal domain in LBP is the primary site for interaction with LPS, including residues Arg119, Lys120, and Lys124 [33]. The docking search space of CD14 was set at the N-terminal hydrophobic pocket functions in order to bind and deliver various lipidated molecules, including LPS [52]. Furthermore, the TLR4/MD-2 complex docking search space was set at the LPS binding site of the 3FXI structure [53]. The ligand structure of EK100 was from ChemSpider (ID: 3657636), and Dexa was from PubChem (CID: 28932). The AutoDock Vina automatically samples different conformations of the ligands to fit the predicted binding site best. The docking results were analyzed and based on the protein–ligand complex’s binding affinity (kcal/mol).

### 4.12. Statistical Analysis

The data were presented as mean ± SEM and analyzed using a one-way analysis of variance (ANOVA). Differences between the LPS-treated and -untreated (control) groups were considered statistically significant at the level of # *p* < 0.05, compared to the negative LPS-stimulated group; * *p* < 0.05, ** *p* < 0.01, and *** *p* < 0.001 compared to the LPS group.

## 5. Conclusions

In conclusion, EK100 suppressed the inflammatory effects via the LPS transfer cascade to the TLR4/MD-2 complex via LBP and CD14, and then, it attenuated the IκB/NF-κB inflammatory transcription factor signaling pathway (Figure 8). EK100 could be a novel strategy to interfere with the LPS/TLR4 signaling pathway to treat inflammatory diseases.

## Figures and Tables

**Figure 1 ijms-22-06511-f001:**
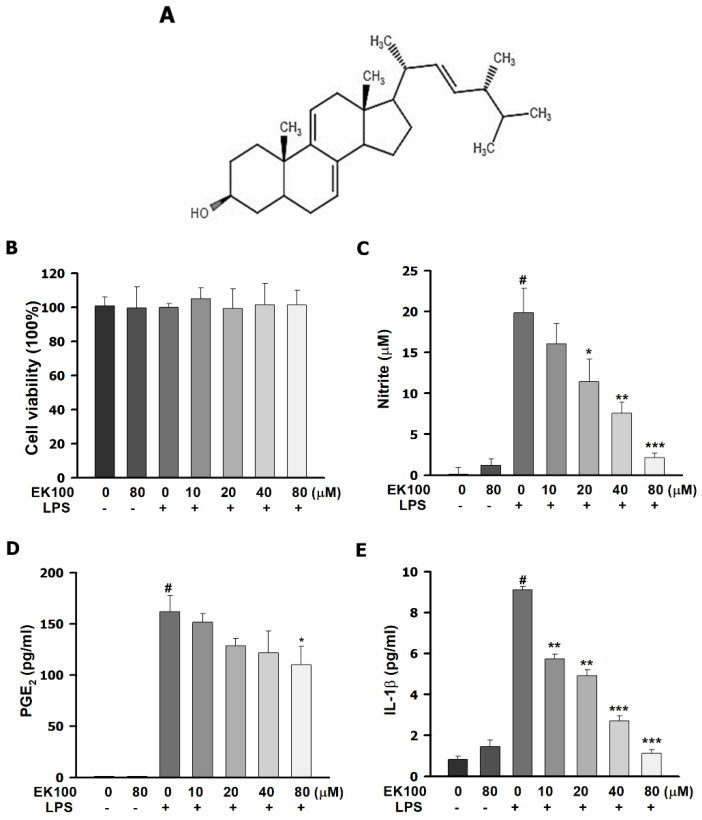
Effects of EK100 on the cell viability and cytokines release in LPS-stimulated RAW264.7 murine macrophage cells. (**A**) The chemical structure of EK100; RAW264.7 cells were pretreated with EK100 (0, 10, 20, 40, and 80 µM) for 1 h, then incubated with LPS (100 ng/mL) for 24 h. The suspension media was separated from the remaining cells. (**B**) In the remaining cells, we analyzed the cytotoxicity with the MTT reagent (*n* = 3). (**C**) In the suspension media, the NO production was detected with the Griess reagents (*n* = 3). The production of PGE_2_ (**D**) and IL-1β (**E**) was measured by using the specific ELISA kit, respectively (*n* = 3). Data are presented as the means ± standard error of the means (SEM). # *p* < 0.05 compared to the control group, * *p* < 0.05, ** *p* < 0.01, and *** *p* < 0.001 compared to the LPS alone group.

**Figure 2 ijms-22-06511-f002:**
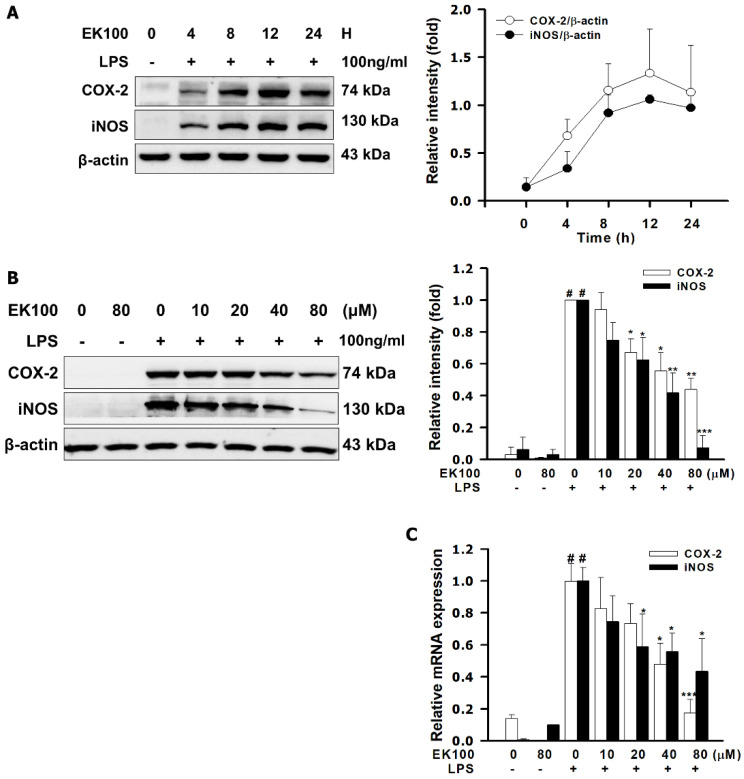
EK100 suppressed LPS-induced mRNA and protein expression of pro-inflammatory mediators of iNOS and COX-2 in RAW264.7 cells. (**A**) Cells were treated with 80 μM of EK100 and then treated with 100 ng/mL LPS at 0, 4, 8, 12, and 24 h to evaluate the protein expression maximum peak of iNOS and COX-2 in LPS-stimulated RAW264.7 cells. (**B**) Cells were treated with 0, 10, 20, 40, and 80 μM of EK100 for 1 h and incubated with 100 ng/mL of LPS for 30 min. The protein expression of iNOS and COX-2 were detected by western blotting (*n* = 3). (**C**) The expression of mRNA of iNOS and COX-2 were detected by Q-PCR (*n* = 3), as designated in the Section 4. Data are presented as the means ± SEM. # *p* < 0.05 compared with the control group, * *p* < 0.05, ** *p* < 0.01, and *** *p* < 0.001 compared with the LPS alone group.

**Figure 3 ijms-22-06511-f003:**
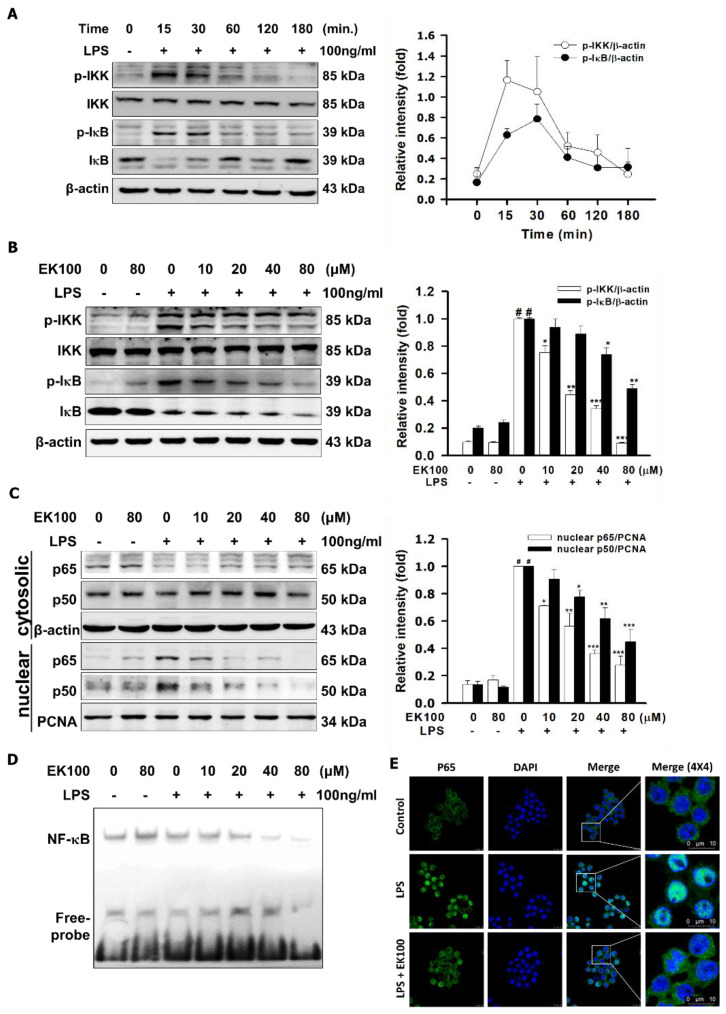
Effects of EK100 on the phosphorylation of IKK/IκB and the translocation of NF-κB in LPS-induced RAW264.7 cells. (**A**) Cells were treated with 80 μM EK100 and then treated with 100 ng/mL LPS at 15, 30, 60, 120, and 180 min to evaluate the protein maximum peak of p-IKK and p-IκB in LPS-stimulated RAW264.7 cells. (**B**) Cells were pretreated with 0, 10, 20, 40, and 80 μM of EK100 for 1 h and stimulated with 100 ng/mL of LPS for 30 min. The protein expression of p-IKK, IKK, p-IκB, and IκB was detected by western blotting (*n* = 3). (**C**) The protein expressions of transcription factor p65 and p50 from the cytoplasm to the nucleus were measured by western blotting (*n* = 3). (**D**) With the EMSA assay, we detected that NF-κB bonded to DNA developed the DNA–NF-κB complex. (**E**) The localization of p65 in the cytoplasm and nucleus were measured by immunofluorescence staining, as designated in the Section 4 (*n* = 3). Data are presented as means ± SEM. # *p* < 0.05 compared with the control group, * *p* < 0.05, ** *p* < 0.01, and *** *p* < 0.001 compared with the LPS group.

**Figure 4 ijms-22-06511-f004:**
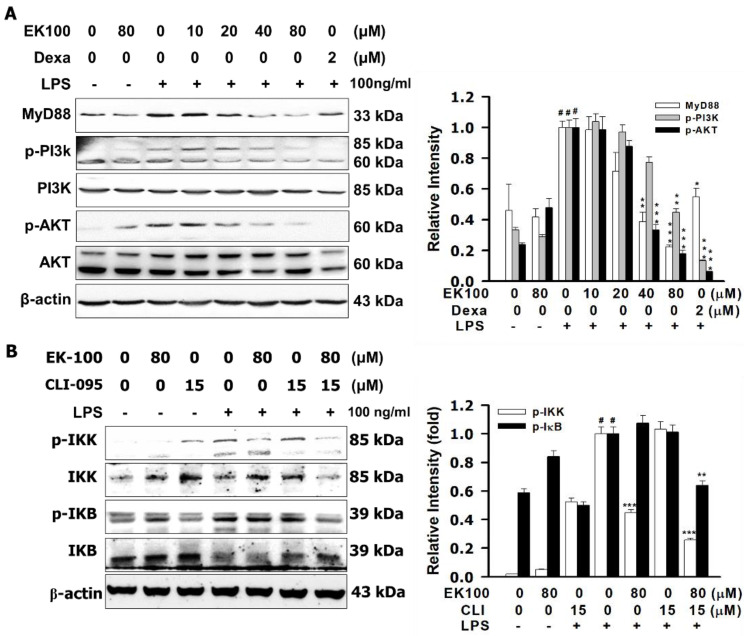
Effects of EK100 on the PI3K/Akt signaling pathway in LPS-induced RAW264.7 cells. (**A**) Cells were pretreated with EK100 (0, 10, 20, 40, and 80 μM) and Dexa (2 μM) and then stimulated with 100 ng/mL of LPS. The phosphorylation of PI3K and Akt were analyzed (*n* = 3). (**B**) The EK100 (80 μM) inhibitor, CLI-095, (15 μM) influenced the LPS-stimulated phosphorylation of IKK and IKB protein expression (*n* = 3). (**C**) EK100 (80 μM) plus TLR4 inhibitor, CLI-095, (15 μM) influenced the LPS-stimulated phosphorylation of p-Akt and MyD88 protein expression (*n* = 3). (**D**) EK100 (80 μM) plus TLR4 inhibitor, CLI-095, (15 μM) influenced LPS-stimulated iNOS and COX-2 protein expression in RAW264.7 cells, and they were analyzed (*n* = 3) as designated in the Section 4. Data are presented as means ± S.E.M. # *p* < 0.05 compared with the control group, * *p* < 0.05, ** *p* < 0.01, and *** *p* < 0.001 compared with the LPS only group.

**Figure 5 ijms-22-06511-f005:**
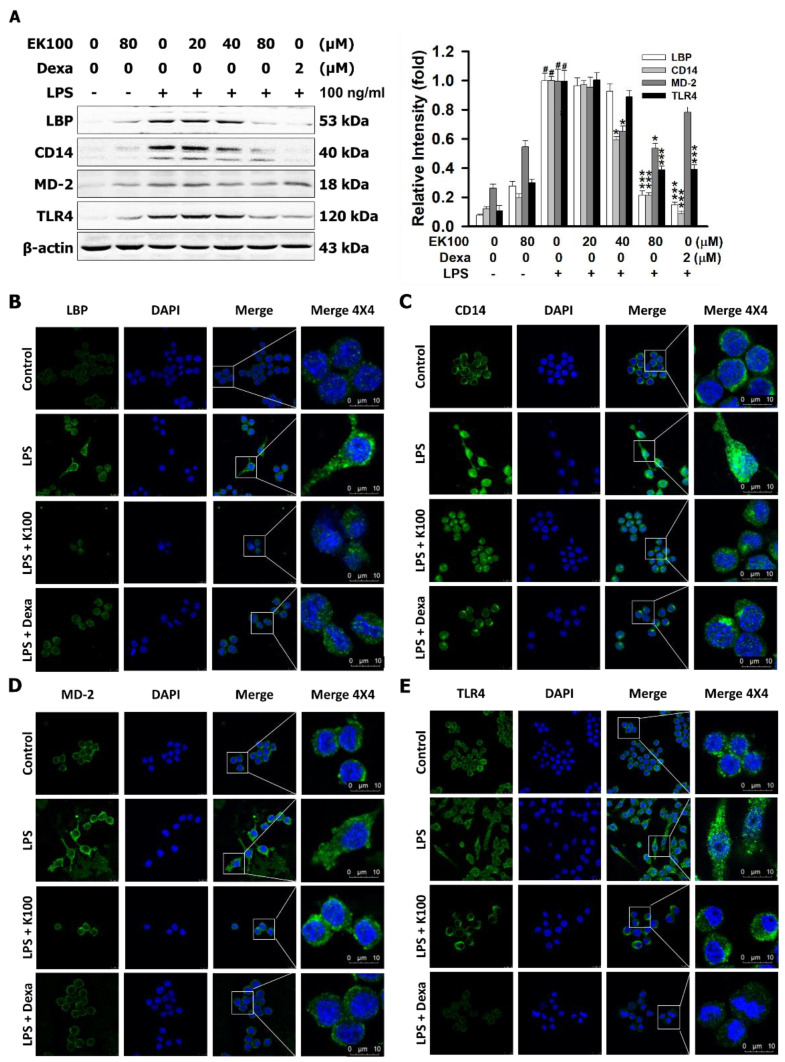
Effects of EK100 on the protein expression of LBP, CD14, and TLR4/MD-2 co-receptors in LPS-stimulated RAW264.7 cells. Cells were pretreated with EK100 (0, 10, 20, 40, and 80 μM) and stimulated with LPS (100 ng/mL) for 24 h. (**A**) The protein expressions of LBP, CD14, MD-2, and TLR4 were measured by western blotting, as designated in the Section 4 (*n* = 3). Representative images were measured with IF staining of (**B**) LBP, (**C**) CD14, (**D**) MD-2, and (**E**) TLR4. All data were presented as the mean ± SEM (*n* = 3). # *p* < 0.05 compared with the control group, * *p* < 0.05 and *** *p* < 0.001 as compared with the LPS alone group.

**Figure 6 ijms-22-06511-f006:**
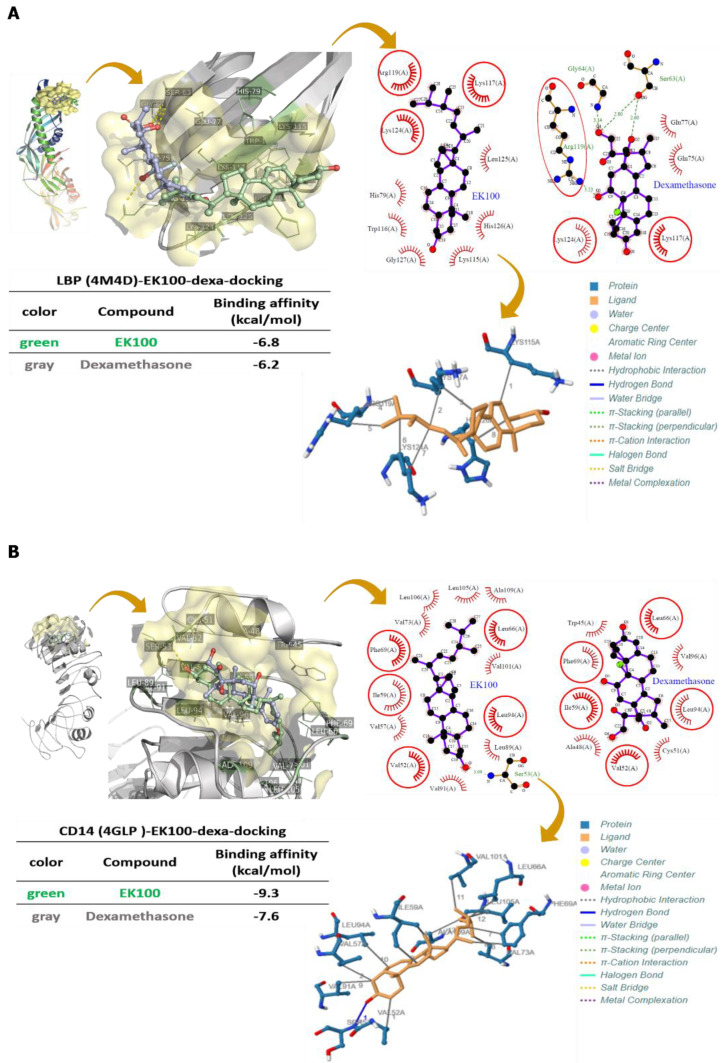
Effects of EK100 on LPS binding to LBP, CD14, and TLR4/MD-2 with protein-ligand docking. (**A**) LBP binding with EK100 and Dexa. The binding affinities of EK100 and Dexa, binding to the LBP protein. (**B**) CD14 binding with EK100 and Dexa. The binding affinities of EK100 and Dexa in the binding of CD14. (**C**) TLR4/MD-2 binding with EK100 and Dexa. The binding affinities of EK100 and Dexa, binding to TLR4/MD-2. These diagrams are shown on the right panel. Red or pink eyebrow-like icons illustrate hydrophobic interactions. These data are displayed at the bottom of the left panel. The binding site is shown as a surface model, and the inhibitor is displayed as a ball and stick model. The green dash line indicates the hydrogen bond pairing with each other. The red circles identify the residues on each plot that were equivalent.

**Figure 7 ijms-22-06511-f007:**
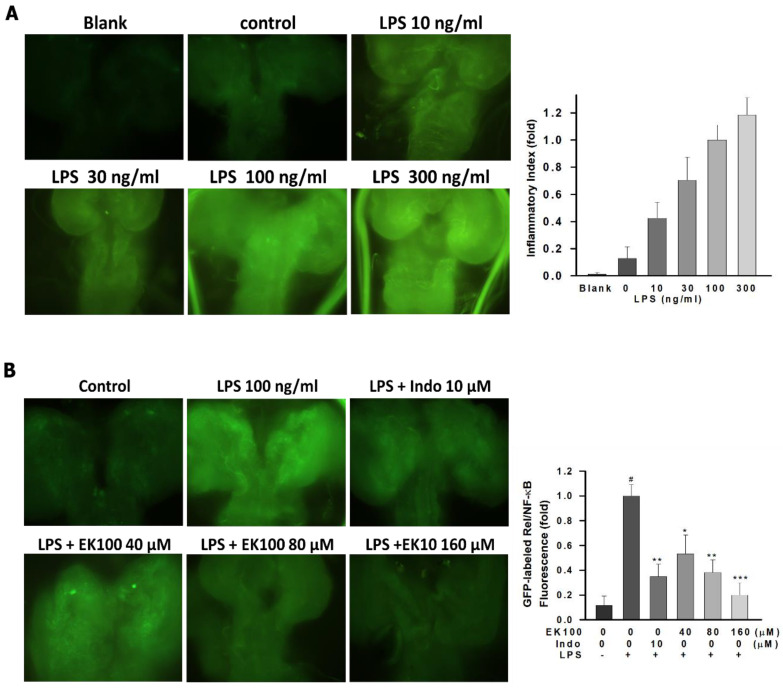
Effects of EK100 on the GFP-labeled Rel/NF-κB fluorescence reporter gene expression in *Drosophila*. (**A**) The GFP-labeled Rel/NF-κB reporter gene showed GFP expression in the brains of larvae *Drosophila*. The leak of the GFP-labeled Rel/NF-κB of larvae in a wild-type genetic background is shown as a blank group. It was pretreated with LPS (10, 30, 100, and 300 ng/mL) for 98 h. The GFP-labeled Rel/NF-κB fluorescence expression was detected using a microscope with a GFP fluorescence channel (Nikon BX51 (Nikon, Tokyo, Japan) with a charge-coupled device (CCD) camera (SPOT; Diagnostic Instruments; Sterling Heights, MI). (**B**) EK100 (0, 40, 80, and 160 μM) or indomethacin (10 μM) were added to *D**rosophila* in a culture medium and incubated with LPS (100 ng/mL) for 96 h. Relative GFP-labeled Rel/NF-κB fluorescence reporter gene expression in whole larvae and the values obtained with dissected tissues were expressed as folds of this value. The histograms correspond to the mean value ± SEM of the three independent experiments. # *p* < 0.05 compared with the control group, * *p* < 0.05, ** *p* < 0.01 and *** *p* < 0.001 as compared with the LPS alone group.

**Figure 8 ijms-22-06511-f008:**
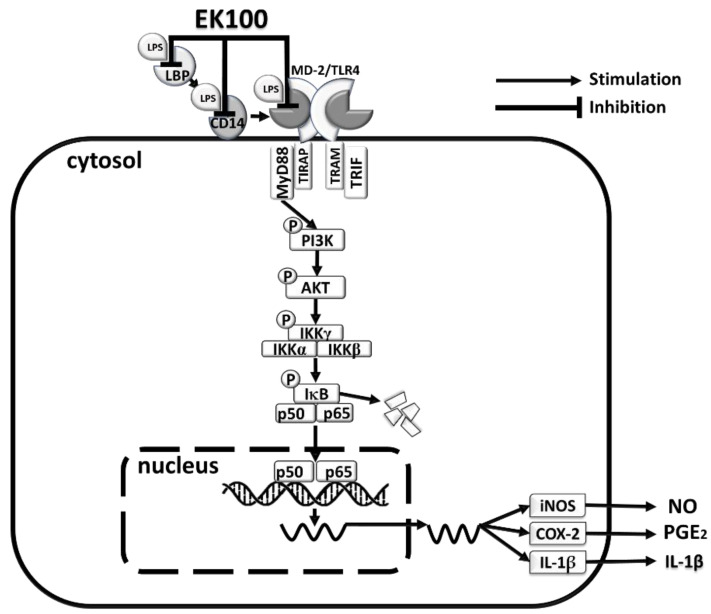
The proposed mechanism represents the anti-inhibiting inflammatory effect of EK100 in LPS-stimulated RAW264.7 cells. Results suggested that EK100 interferes with LPS binding to LBP, CD14, and the TLR4/MD-2 co-receptor, and its activation inhibits TLR4/MyD88-dependent signaling, the phosphorylation of PI3K/AKT/IKK pathways, and the activation of NF-κB signal pathway in LPS-stimulated macrophage-like RAW264.7 cells.

## Data Availability

Not applicable.

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
