# Peer review of "Ergosta-7, 9 (11), 22-trien-3β-ol Interferes with LPS Docking to LBP, CD14, and TLR4/MD-2 Co-Receptors to Attenuate the NF-κB Inflammatory Pathway In Vitro and Drosophila"

_ijms, 2021, doi:10.3390/ijms22126511_

Round 1
Reviewer 1 Report
I am not an expert in experimental studies. As far as the computational study is concern, it is supportive to the manuscript and are presented well.
Author Response
Thank you for your comments and support of our manuscript.

Reviewer 2 Report
This study showed the Ergosta-7, 9 (11), 22-trien-3β-ol (EK100) might alter LPS docking to LBP, CD14, and TLR4/MD-2 co-receptors then suppresses the TLR4/NF-κB inflammatory pathway in LPS-stimulated RAW264.7 cells and Drosophila. Also, this paper found the EK100 can be inhibited the GFP-labeled NF-κB/Rel Fluorescence expression in Drosophila. Therefore, I would like to suggest to be accepted this manuscript after simple minor revision in IJMS.
- In paper, Ergosta-7, 9 (11), 22-trien-3β-ol was designated as EK100. Is there any special reason for this?
- Recommended the revision: ml → mL
- In Figure 1. “* p < 0.05, ** p < 0.01, and *** p < 0.001 were compared to the LPS alone group.”: ‘were’ should be deleted, because of unify with Figure 2-5.
Author Response
- In paper, Ergosta-7, 9 (11), 22-trien-3β-ol was designated as EK100. Is there any special reason for this?
Response 1: Thank you for the kindful comments. To accessible name in the manuscript, therefore the author of Yueh-Hsiung Kuo's team give a short name EK100 for Ergosta-7, 9 (11), 22-trien-3β-ol throughout the whole manuscript. EK100 was a known bioactive ergostatrien compound isolated from the Antrodia camphorate by Professor Yueh-Hsiung Kuo [1]. Professor Kuo found that EK100 has several biological effects and could develop a novel natural anti-inflammatory medicine.
- According to the previous studies, EK100 showed inhibit the pro-inflammatory cytokine expression in RAW264.7 cells [2], anti-inflammation in chronic-alcohol fed mice [3], anti-inflammatory and anti-photodamaging effects on hairless mouse skin [4], attenuated oxidative stress, inflammation, and liver injury in vitro and in vivo [5], downregulated the expression of p65NF-κ-B and caspase 3, and activated the Akt/GSK3/catenin-associated neurogenesis in an acute ischemic stroke (AIS) murine model [6].
- Moreover, EK100 inhibited the diabetes and the hyperlipidemia in high-fat-diet treated mice [1].
- EK100 also alleviates intracerebral hemorrhage-induced brain injury in mice and BV-2 microglial activation [7].
- Foremore, EK100 presented the anti-fatigue properties through the preservation of energy storage, increasing blood glucose and liver glycogen content, and decreasing the serum levels of lactate, ammonia, BUN, and CK in exercise performance of mice [8].
In summary, EK100, an ergostatrien, was found several excellent bioactive effects by Professor Kuo's research team. Thus, Dr. Kuo designated it as EK100.
- Recommended the revision: ml → mL
Response 2: Thanks for your suggestion. We have already corrected it, which can be seen in the revised manuscript.
- In Figure 1. “* p < 0.05, ** p < 0.01, and *** p < 0.001 were compared to the LPS alone group.”: ‘were’ should be deleted, because of unify with Figure 2-5.
Response 3: Thanks for your suggestion. We have already deleted "were" which can be seen in the revised manuscript.
References
- Kuo, Y.H.; Lin, C.H.; Shih, C.C. Ergostatrien-3β-ol from Antrodia camphorata inhibits diabetes and hyperlipidemia in high-fat-diet treated mice via regulation of hepatic related genes, glucose transporter 4, and AMP-activated protein kinase phosphorylation. J Agric Food Chem 2015, 63, 2479-2489, doi:10.1021/acs.jafc.5b00073.
- Kao, S.T.; Kuo, Y.H.; Wang, S.D.; Hong, H.J.; Lin, L.J. Analogous corticosteroids, 9A and EK100, derived from solid-state-cultured mycelium of Antrodia camphorata inhibit pro-inflammatory cytokine expression in macrophages. Cytokine 2018, 108, 136-144, doi:10.1016/j.cyto.2018.03.035.
- Chang, Y.Y.; Liu, Y.C.; Kuo, Y.H.; Lin, Y.L.; Wu, Y.S.; Chen, J.W.; Chen, Y.C. Effects of antrosterol from Antrodia camphorata submerged whole broth on lipid homeostasis, antioxidation, alcohol clearance, and anti-inflammation in livers of chronic-alcohol fed mice. Journal of ethnopharmacology 2017, 202, 200-207, doi:10.1016/j.jep.2017.03.003.
- Kuo, Y.H.; Lin, T.Y.; You, Y.J.; Wen, K.C.; Sung, P.J.; Chiang, H.M. Antiinflammatory and Antiphotodamaging Effects of Ergostatrien-3β-ol, Isolated from Antrodia camphorata, on Hairless Mouse Skin. Molecules 2016, 21, doi:10.3390/molecules21091213.
- Chao, T.Y.; Hsieh, C.C.; Hsu, S.M.; Wan, C.H.; Lian, G.T.; Tseng, Y.H.; Kuo, Y.H.; Hsieh, S.C. Ergostatrien-3β-ol (EK100) from Antrodia camphorata Attenuates Oxidative Stress, Inflammation, and Liver Injury In Vitro and In Vivo. Prev Nutr Food Sci 2021, 26, 58-66, doi:10.3746/pnf.2021.26.1.58.
- Wang, Y.H.; Chern, C.M.; Liou, K.T.; Kuo, Y.H.; Shen, Y.C. Ergostatrien-7,9(11),22-trien-3β-ol from Antrodia camphorata ameliorates ischemic stroke brain injury via downregulation of p65NF-κ-B and caspase 3, and activation of Akt/GSK3/catenin-associated neurogenesis. Food Funct 2019, 10, 4725-4738, doi:10.1039/c9fo00908f.
- Hsueh, P.J.; Wang, M.H.; Hsiao, C.J.; Chen, C.K.; Lin, F.L.; Huang, S.H.; Yen, J.L.; Tsai, P.H.; Kuo, Y.H.; Hsiao, G. Ergosta-7,9(11),22-trien-3β-ol Alleviates Intracerebral Hemorrhage-Induced Brain Injury and BV-2 Microglial Activation. Molecules 2021, 26, doi:10.3390/molecules26102970.
- Chen, Y.M.; Sung, H.C.; Kuo, Y.H.; Hsu, Y.J.; Huang, C.C.; Liang, H.L. The Effects of Ergosta-7,9(11),22-trien-3β-ol from Antrodia camphorata on the Biochemical Profile and Exercise Performance of Mice. Molecules 2019, 24, doi:10.3390/molecules24071225.

Reviewer 3 Report
see attached file

Author Response
Comments and Suggestions for Authors
This is an interesting paper further describing the anti-inflammatory activity of EK100, particularly its effect on the LPS/TLR4 signalling pathway. I would recommend this manuscript for publication in IJMS following the following minor corrections:
- A thorough English language is required regarding grammar and usage. I believe that the publisher offers this service.
Response 1: Thanks for your suggestion. We have double-check and correct grammar throughout the whole manuscript, which can be seen in the revised manuscript.
- Abbreviations CLI-095 and GFP: The authors have generally done a good job of defining abbreviations the first time they are used. However, the abbreviations CLI-095 and GFP are not defined in the manuscript.
Response 2: Thanks for your suggestion. We have already corrected them, which can be seen in the revised manuscript.
- Following on from this, the paper would benefit from a dedicated abbreviations section. I believe this is a mandatory section of IJMS, but this seems to have been omitted from this paper.
Response 3: Thanks for your suggestion. We have already created an abbreviations section, which can be seen in the revised manuscript.
- Nitric oxide radical vs nitrite: In Section 2.1 and 4.4, the authors seem to imply that nitrite is the cause of oxidative damage. More clarity is needed to explain that nitric oxide radical is in fact the cause of oxidative damage and the species of interest, whereas nitrite is a metabolite of the nitric oxide radical, and therefore the detection of nitrite is used to estimate the concentration of nitric oxide radical.
Response 4: Thanks for your suggestion. Oxygen-derived free radicals and related oxidants are ubiquitous and short-lived intermediates formed in aerobic organisms. These reactive species participate in redox reactions leading to oxidative modifications in biomolecules, among which proteins and lipids are preferential targets. Despite a broad array of enzymatic and nonenzymatic antioxidant systems in mammalian cells and microbes, excess oxidant formation causes the accumulation of new products that may compromise cell function and structure, leading to cell degeneration and death [1]. Notably, physiological levels of oxidants also modulate cellular processes via homeostatic redox-sensitive cell signaling cascades. However, nitric oxide (•NO), a free radical and weak oxidant represent a master physiological regulator via reversible interactions with heme proteins. The bioavailability and action of •NO is modulated by its fast reaction with superoxide radical (O∙−2O2•−), which yields an unusual and reactive peroxide, peroxynitrite, representing the merging of the oxygen radicals and •NO pathways [2].
Moreover, the biochemical analyses show that early killing of bacteria by macrophages coincides by nitric oxide, superoxide anion (O2·−), hydrogen peroxide (H2O2), and peroxynitrite (ONOO−) production, which nitric oxide was invole the context of antipathogen activity and immune regulation [3,4]. However, otherwise, NO is involved in many physiological processes, including blood pressure, immune response, and neural communication. Therefore its accurate detection and quantification is critical to understanding health and disease. [5], [6]. Nitric oxide (NO) is a critical signaling molecule marked by complex chemistry and varied biological responses depending on the context of the redox environment. NO can contribute to tissue injury in inflammation and be causative of oxidative damage and signal as an adaptive molecule to limit inflammatory signaling in multiple cell types and tissues [7] [8]. Nitrite and nitrate signaling to be critical in regulating inflammation in various conditions and have been shown to protect against tissue injury[9]. The stability of nitrate and nitrite compared with NO and the ability to deliver systemically and perhaps induce a local release of NO at a site where the biology favors the reduction to NO have created a renaissance of interest in NO-based therapies disease [7]. The effects of nitrate/nitrite and NO are diverse. Perhaps the most direct and best-described pathway involves binding NO to soluble guanylate cyclase (sGC), leading to cGMP generation, activation of cGMP-dependent kinase pathways, and an ultimate myriad of effects.
However, NO, a free radical and weak oxidant, represents the master physiological regulator and an essential role in inflammatory biological responses. Therefore its accurate detection and quantification are critical to understanding health and disease. There are many quantification methods of NO metabolites in biological samples that provide valuable information with regards to in vivo NO production, bioavailability, and metabolism. The Griess reaction is a two-step diazotization reaction in which could accurately detect the NO-derived nitrosating agent, dinitrogen trioxide (N2O3) generated from the acid-catalyzed formation of nitrous acid from nitrite (or autoxidation of NO), reacts with sulfanilamide to produce a diazonium ion which is then coupled to N-(1-naphthyl)ethylenediamine to form a chromophoric azo product that absorbs strongly at 540 nm [10].
In summary, nitric oxide radicals could be the cause of oxidative damage in tissue. However, it also shows that the early killing of bacteria by macrophages symbolizes inflammatory and immune responses. The Griess reagent method available for detecting nitrite is used to estimate the concentration of nitric oxide radicals. Most commonly used allow accurate and sensitive quantification of NO products/metabolites in multiple biological matrices under normal physiological conditions.
- Axis labels could be less cluttered: The axis labels of the graphs in Figures 1-5 use an unnecessarily 'elongated' font which makes the labels more difficult to read. I therefore recommend changing the font of the axes.
Response 5: Thanks for your suggestion. We have already corrected it, which can be seen in the revised manuscript.
- Stereochemistry of the compound EK100: In Figure 1a, the drawn structure is missing the assignment of stereochemistry at two stereocentres, leaving the reader unsure if the tested material is a single isomer, or a mixture of (some or all of) the four possible isomers shown below: The authors should therefore clarify the identity of the isomer obtained from their commercial source, if known, in Section 4.1. Or, if the stereo-identity at these positions is unknown, or a mixture of isomers was studied, then the authors should state this in Section 4.1
Response 6: Thanks for your comments. In Figure 1a and Figure 2, both ring junctions between A/B and C/D are trans. Otherwise, the methyl groups attached on C-10 and C-13 would resonate much more downfield on the proton NMR spectrum. Thus, both H-5 and H-14 are α-oriented (Figures 3A and 3B). The stereochemistry of EK100 could be assigned as follow molecular structure (Figure 2). HPLC spectra showed below (Figure 4).
Figure 1. (a) EK100 structure of IJMS-1225299ˉ
Figure 2. EK100 molecular structure identification.
AB
Figure 3. EK100 NMR spectra. A. H NMR spectra. B. C13 NMR spectra.
- HPLC Elution solvents conditions:
|
Time(min) |
H2O (%, v/v) |
Acetonitrile(%, v/v) |
|
0 |
60 |
40 |
|
50 |
5 |
95 |
|
60 |
5 |
95 |
- Detector: UV/visible Detector 280nm
- HPLC spectra
Figure 4. EK100 HPLC spectra
References
- Gerschman, R.; Gilbert, D.L.; Nye, S.W.; Dwyer, P.; Fenn, W.O. Oxygen Poisoning and X-irradiation: A Mechanism in Common. Science 1954, 119, 623-626, doi:10.1126/science.119.3097.623.
- Radi, R. Oxygen radicals, nitric oxide, and peroxynitrite: Redox pathways in molecular medicine. Proceedings of the National Academy of Sciences 2018, 115, 5839-5848, doi:10.1073/pnas.1804932115.
- Vazquez-Torres, A.; Jones-Carson, J.; Mastroeni, P.; Ischiropoulos, H.; Fang, F.C. Antimicrobial actions of the NADPH phagocyte oxidase and inducible nitric oxide synthase in experimental salmonellosis. I. Effects on microbial killing by activated peritoneal macrophages in vitro. J Exp Med 2000, 192, 227-236, doi:10.1084/jem.192.2.227.
- Wink, D.A.; Hines, H.B.; Cheng, R.Y.; Switzer, C.H.; Flores-Santana, W.; Vitek, M.P.; Ridnour, L.A.; Colton, C.A. Nitric oxide and redox mechanisms in the immune response. J Leukoc Biol 2011, 89, 873-891, doi:10.1189/jlb.1010550.
- Bor-Kucukatay, M.; Wenby, R.B.; Meiselman, H.J.; Baskurt, O.K. Effects of nitric oxide on red blood cell deformability. Am J Physiol Heart Circ Physiol 2003, 284, H1577-1584, doi:10.1152/ajpheart.00665.2002.
- Kubes, P.; Suzuki, M.; Granger, D.N. Nitric oxide: an endogenous modulator of leukocyte adhesion. Proc Natl Acad Sci U S A 1991, 88, 4651-4655, doi:10.1073/pnas.88.11.4651.
- Waltz, P.; Escobar, D.; Botero, A.M.; Zuckerbraun, B.S. Nitrate/Nitrite as Critical Mediators to Limit Oxidative Injury and Inflammation. Antioxid Redox Signal 2015, 23, 328-339, doi:10.1089/ars.2015.6256.
- Dezfulian, C.; Shiva, S.; Alekseyenko, A.; Pendyal, A.; Beiser, D.G.; Munasinghe, J.P.; Anderson, S.A.; Chesley, C.F.; Vanden Hoek, T.L.; Gladwin, M.T. Nitrite therapy after cardiac arrest reduces reactive oxygen species generation, improves cardiac and neurological function, and enhances survival via reversible inhibition of mitochondrial complex I. Circulation 2009, 120, 897-905, doi:10.1161/circulationaha.109.853267.
- Alef, M.J.; Vallabhaneni, R.; Carchman, E.; Morris, S.M., Jr.; Shiva, S.; Wang, Y.; Kelley, E.E.; Tarpey, M.M.; Gladwin, M.T.; Tzeng, E., et al. Nitrite-generated NO circumvents dysregulated arginine/NOS signaling to protect against intimal hyperplasia in Sprague-Dawley rats. J Clin Invest 2011, 121, 1646-1656, doi:10.1172/jci44079.
- Grisham, M.B.; Johnson, G.G.; Lancaster, J.R., Jr. Quantitation of nitrate and nitrite in extracellular fluids. Methods Enzymol 1996, 268, 237-246, doi:10.1016/s0076-6879(96)68026-4.
